# Label-Free Detection of Human Coronaviruses in Infected Cells Using Enhanced Darkfield Hyperspectral Microscopy (EDHM)

**DOI:** 10.3390/jimaging8020024

**Published:** 2022-01-25

**Authors:** Devadatta Gosavi, Byron Cheatham, Joanna Sztuba-Solinska

**Affiliations:** 1Department of Biological Sciences, Auburn University, 120 W. Samford Ave, Rouse Life Sciences Building, Auburn, AL 36849, USA; dag0042@auburn.edu; 2Cytoviva, Inc., 570 Devall Drive Suite 301, Auburn, AL 36832, USA; byron.cheatham@cytoviva.com

**Keywords:** human coronavirus (HCoV), plaque assay, enhanced darkfield hyperspectral microscopy (EDHM)

## Abstract

Human coronaviruses (HCoV) are causative agents of mild to severe intestinal and respiratory infections in humans. In the last 15 years, we have witnessed the emergence of three zoonotic, highly pathogenic HCoVs. Thus, early and accurate detection of these viral pathogens is essential for preventing transmission and providing timely treatment and monitoring of drug resistance. Herein, we applied enhanced darkfield hyperspectral microscopy (EDHM), a novel non-invasive, label-free diagnostic tool, to rapidly and accurately identify two strains of HCoVs, i.e., OC43 and 229E. The EDHM technology allows collecting the optical image with spectral and spatial details in a single measurement without direct contact between the specimen and the sensor. Thus, it can directly map spectral signatures specific for a given viral strain in a complex biological milieu. Our study demonstrated distinct spectral patterns for HCoV-OC43 and HCoV-229E virions in the solution, serving as distinguishable parameters for their differentiation. Furthermore, spectral signatures obtained for both HCoV strains in the infected cells displayed a considerable peak wavelength shift compared to the uninfected cell, indicating that the EDHM is applicable to detect HCoV infection in mammalian cells.

## 1. Introduction

Viruses belonging to different families pose a continuous threat to public health and worldwide stability. The unprecedented outbreaks of severe acute respiratory syndrome-associated coronavirus (SARS-CoV) [1], Middle East respiratory syndrome coronavirus (MERS-CoV) [2], and the ongoing pandemic of severe acute respiratory syndrome-associated coronavirus 2 (SARS-CoV-2) [3] poised human coronaviruses to take prominence on the world stage, highlighting the necessity for readily available, accurate, and fast diagnostic testing methods.

Coronaviruses are positive-sense, single-stranded RNA viruses that belong to the family *Coronaviridae*, which includes four genera, several sub-genera, and species. Four human coronaviruses (HCoVs) are globally endemic, including two *Alphacoronavirus* (HCoV-229E and HCoV-NL63) and two *Betacoronavirus* (HCoV-OC43 and HCoV-HKU1). These HCoVs account for 10% to 30% of upper respiratory tract infections in adults [4]. The emergence of HCoV-229E was estimated to occur about 200 years ago, and it involved a transfer from bats to alpacas and then to humans. About 120 years ago, HCoV-OC43, which was considered to have a common ancestor with bovine coronavirus, was transmitted from cattle to humans [5]. Subsequently, both HCoV-229E and HCoV-OC43 were isolated from the nasal cavities of people with the common cold in the 1960s [6]. In the 1970s, studies used serology and viral culture linked HCoVs-229E and -OC43 with 8% of lower respiratory tract infection cases in hospitalized infants [7]. HCoV-NL63 and HCoV-HKU1 were discovered in the early 2000s from persons with bronchiolitis and pneumonia [8,9]. In 2002 a *Betacoronavirus* originating in bats spread from civets to humans, causing severe respiratory disease and taking the name SARS-CoV [10]. In 2012 a *Betacoronavirus* spread from camels to humans in Saudi Arabia, causing a similar clinical syndrome as SARS, taking the name MERS-CoV [11]. The latest SARS-CoV-2 is closely related to *Betacoronaviruses* detected in bats (88% sequence identity) but is genetically distinct from SARS-CoV (~79% sequence identity) [12]. It is understood that a broad host range [13], frequent cross-species transmission, stability of the virus in the environment, and changes in the tissue tropism [14], support the emergence of new highly infectious human coronaviruses.

There is a general lack of adequate clinical characteristics and epidemiological data on human coronaviruses. This knowledge gap primarily persists due to the difficulties in developing cellular and animal model systems to uncover the mechanisms of viral pathogenicity and designing adequate therapeutic strategies [15]. Also, a critical deficiency lies in the available diagnostic tests that allow for early, rapid, and sensitive detection of coronaviruses. Quantitative reverse transcriptase-polymerase chain reaction (RT-qPCR) is currently the most frequently used technique to detect viral nucleic acids present in bodily fluids. Testing facilitates the prevention of spread between persons and communities, including asymptomatic infected people, whose viral shedding can unintentionally spread the infection to the elderly and immunocompromised. However, the total turnaround time for RT-qPCR can exceed two days, and the results can be burdened with false-negative output caused by a low viral load or the presence of genetic variants of the virus [16] and false-positive results arising from cross-contamination [16]. Serological testing complements virus detection, indicating past infection, which could be harnessed for therapeutic gain [17]. Here, antibodies are detected by enzyme-linked immunosorbent assay using a qualitative detection of IgG or IgM antibodies [18]. Such tests determine an immune response against the viral antigen and may be helpful to assess protection against subsequent viral exposure and for contact tracing purposes. However, serological tests usually detect the antibodies, not the virus itself, and the effectiveness of these assays during the early stages of infection when adaptive immunity is building is, therefore, limited [19]. Fluorescence microscopy is another widely employed detection method. It allows for the visualization of viral particles and is one of the main techniques to study molecular aspects of the viral infectivity cycle and virus interactions within the complex biological environment in a minimally invasive manner [20]. However, sample preparation and analysis are time-consuming, as they require the labeling of viruses with fluorescent dyes [21]. Also, endogenous cellular autofluorescence can mask a proper fluorescence signal, limiting the ability to visualize low abundance fluorescently labeled virions [22]. In response to the limitations of the above-mentioned diagnostic methods, new platforms are actively being pursued.

This manuscript outlines the advantages of using the enhanced darkfield hyperspectral microscopy (EDHM) as a novel tool that allows for label-free detection of two HCoV strains, i.e., OC43 and 229E, in infected mammalian cells. The technology (developed by CytoViva, Inc., Auburn, AL, USA) offers high-contrast visualization of low-contrast objects that are typically not visible by conventional bright-field microscopy. These factors are of great interest for the microscopic examination of biological samples, as the technology does not require the application of contrast agents. When combined with hyperspectral imaging, the optical image with spectral and spatial details can be acquired in a single measurement without direct contact between the specimen and the sensor [23]. Furthermore, the integrated system can directly map spectral signatures specific for a given viral strain in a complex biological milieu based on its unique optical spectrum. Accordingly, this method is sensitive to subtle spectral changes, ensuring thorough discrimination of biological entities or changes in tracking agents over time [24]. As such, EDHM represents a novel alternative to the available diagnostic tools for detecting various infectious agents.

## 2. Materials and Methods

### 2.1. Cell Lines and Culture Conditions

Human colon adenocarcinoma (HCT-8 cells, ATCC CCL-244) were seeded in 75 cm^2^ cell culture flasks at 4.7 × 10^6^ cells/mL and grown in RPMI-1640 medium (ATCC 30-2001) supplemented with 10% horse serum (HS) (Fisher scientific, SH3007403, Waltham, MA, USA) and 1x Penicillin/Streptomycin/Glutamine (Gibco, 10378016) at 37 °C in 5% CO_2_. Medical research council cells strain 5 (MRC-5 cells, ATCC CCL-171) were seeded in 75 cm^2^ cell culture flasks at 1 × 10^6^ cells/mL and grown in Eagle’s Minimum Essential Medium (EMEM, ATCC 30-2003) supplemented with 10% fetal bovine serum (FBS) (ATCC 30-2020), 1x Penicillin/Streptomycin/Glutamine (Gibco, 10378016) at 37 °C in 5% CO_2_. The monolayers of both HCT-8 or MRC-5 cells were washed with 1x Phosphate-Buffered Saline (PBS, Corning 21-040-CM) and treated with Trypsin-EDTA solution (VWR L0154-0100) for 5 min at 37 °C in 5% CO_2_ to set up subculture every 2–3 days.

### 2.2. Coronavirus Propagation

HCoV-OC43 and HCoV-229E viral stocks were originally obtained from ATCC (VR-1558 and VR-740, respectively). The 80–90% confluent HCT-8 cells (~6 × 10^6^ cells/mL) were infected with HCoV-OC43 at a viral titer of 1.12 × 10^5^ PFU/mL (multiplicity of infection, MOI 0.05). The 80–90% confluent MRC-5 cells (~5.4 × 10^6^ cells/mL) were transfected with HCoV-229E at a viral titer of 1.12 × 10^6^ PFU/mL (MOI 0.5). The viral adsorption was performed in a minimum volume of serum-free media to facilitate the efficient diffusion of viral particles into the target cells. The HCT-8 cells were incubated for 1 h at 33 °C, while MRC-5 cells were incubated for 1 h at 35 °C with continuous rocking. Viral suspensions were aspirated, and RPMI-1640 medium containing 2% HS was added to HCoV-OC43-infected cell culture and incubated for 11 days at 33 °C in 5% CO_2_, while HCoV-229E-infected cells were incubated with EMEM medium containing 2% FBS media for 5 days at 35 °C in 5% CO_2_. Cell morphology was monitored daily by bright-field microscopy.

### 2.3. Preparation of Viral Stocks

Following 11 or 5 days post-infection for HCT-8 and MRC-5 cell lines, respectively, the cells were harvested, and virions were released by four freeze–thaw cycles in respective maintenance media (serum-free). The suspensions were clarified by centrifugation at 1500× *g* for 15 min at 0 °C, and the supernatant was mixed with 8.3% polyethylene glycol (PEG, VWR 97061-102) with rapid mixing on ice. The solution was incubated at +4 °C overnight, followed by centrifugation at 1800× *g* for 30 min at 0 °C. The virus pellets were resuspended in 500 µL NET buffer (0.15 M NaCl, 0.0005 M EDTA, 0.02 M Tris, pH 7.2) and stored at −80 °C.

### 2.4. Plaque Assays

1 × 10^6^ cells (HCT-8 and MRC-5) were seeded in a 6-well cell culture plate to achieve 80–90% confluency. The monolayers were washed twice with 1x PBS. Viral samples were diluted 10-fold in respective maintenance media (serum-free), and the appropriate dilution was added to the corresponding wells. A single well containing uninfected control cells was maintained to monitor cellular viability and plaque formation. The viral adsorption was carried out for 1 h at 33 °C for HCoV-OC43 and 35 °C for HCoV-229E in 5% CO_2_ with the intermittent rocking of the plates for 20 min to ensure even virions adsorption and to prevent the monolayer from drying. The overlay medium was prepared by combining 0.6% agarose (VWR Life Science, 97062-244, Radnor, PA, USA) with respective maintenance media. 2 mL of overlay medium was added to each well, and the plates were incubated for 11 days at 33 °C in 5% CO_2_ for HCoV-OC43 infected cells or 5 days at 35 °C in 5% CO_2_ for HCoV-229E infected cells (Figure 1).

### 2.5. Fixation and Staining

After incubation, 200 µL of 4% paraformaldehyde in a 0.1 M cacodylate buffer solution (Fischer Scientific, Hampton, NH, USA, 50-190-1157) was applied directly to each well, followed by incubation for 1 h at room temperature. The fixing solution was aspirated, and agarose plugs were removed. The cells were stained using 1% crystal violet solution (Millipore Sigma, V5265, Burlington, MA, USA) for 10 min with gentle rocking. The crystal violet solution was removed, and plates were dried to estimate the plaques count. The virus titer was determined according to the following formula: Virus titer (pfu/mL) = the average number of plaques/(dilution factor ∗ volume of diluted virus added to the well).

### 2.6. Preparation of Cell and Viral Samples for EDHM

0.17 mm glass coverslips (Fischer Scientific, 10474379) were coated with 800 µL gelatin (Millipore Sigma, G1393) and incubated for 2 h at room temperature. The gelatin was aspirated, and coverslips were washed with 1x PBS and dried for 3 h at room temperature. The 24-well cell culture plate was seeded at ~5 × 10^5^ cells/mL (MRC-5 or HCT-8) in the respective maintenance medium. 24 h later, the cells were washed with 1x PBS, fixed with 4% 800 µL paraformaldehyde in 0.1 M cacodylate buffer solution and incubated for 15 min at room temperature. The cells were washed with 1x PBS twice before mounting the slides’ coverslips. For the preparation of the virus samples, 5 µL of the viral stock solution (1.12 × 10^5^ pfu/mL of HCoV-OC43 or 1.12 × 10^6^ pfu/mL of HCoV-229E) was incubated with 5 µL 4% formaldehyde in 0.1 M cacodylate buffer solution for 15 min at room temperature and mixed with 6 μL mounting media (50% glycerol, 0.1 M Tris, pH 8.5). The viral samples were placed on a glass slide (Fischer Scientific, 12-549-3) with a coverslip and directed to dark field microscopy analysis.

### 2.7. EDHM Imaging of HCoV Viral Samples

The stock or PEG-precipitated viral samples (HCoV-OC43 or HCoV-229E) were visualized using a CytoViva EDHM imaging system mounted on a standard research-grade optical microscope with an oil immersion darkfield illuminator, which was adjusted in the X, Y, and Z focal range using a 10x objective and a condenser. The optical images of viral samples were examined using a 60x oil adjustable iris objective with the numerical aperture (NA) set to 1.2 NA and the tungsten halogen light source adjusted to 150 watts power. The hyperspectral images were captured with 2x binning of the camera with an exposure of 0.4 s for the OC43 sample and 0.7 s for the 229E sample, producing a total image pixel resolution of 700 × 300 for each sample. The image analysis of each pixel was performed using the CytoViva software (ENVI 4.8). In particular, the hyperspectral image analysis particle filter feature was utilized to identify the viral particles in each sample. The pixel-level spectrums obtained from identified particles were grouped as a region of interest (ROI). The mean spectral analysis algorithm was then employed to create a mean spectral response from these ROIs. The mean spectrum was normalized to a peak intensity of one for comparative analysis using a preset spectral math equation built into the spectral analysis software.

### 2.8. EDHM Imaging of Mammalian Cells Infected with HCoV

EDHM images for mammalian cells infected with HCoV-OC43 or HCoV-229E were examined using a 60x oil iris objective with the numerical aperture set to 1.2 NA and the quartz halogen light source adjusted to 150 watts power. The hyperspectral images were captured with an exposure of 0.25 s and 2x binning of the camera. After scanning the entire area of each infected cell, different areas containing suspected viral particles were captured with EDHM, and hyperspectral data cubes of the same size areas were captured for spectral mapping. Here, hyperspectral data cubes are hyperspectral images, which are analyzed by Spectral Angle Mapper (SAM) a robust algorithm utilized to determine the spectral resemblance between two spectral profiles and to match the pixels to the reference spectrum. Both uninfected (negative control) and infected cells were imaged following the same steps.

For hyperspectral mapping analysis, the infected cells were scanned for large areas of individual pixel level spectrum and were grouped in regions of interest (ROIs). Each of these ROIs was then converted to spectral libraries (SLs) representing thousands of individual pixel spectral data obtained from each pixel of ROI. Individual SLs obtained from the infected cells were then filtered against the SLs acquired from the uninfected cells, using the Filter Spectral Library algorithm to remove duplicative spectra and create a Reference Spectral Library (RSL). The RSL was then used to map against all the data cubes obtained from the infected cells for detecting the areas with the same spectral profile as the RSL using the SAM function in the ENVI 4.8 software. The mapped areas were merged in one color and overlaid on the original data cube to observe where the viral particles of interest were located.

## 3. Results

### 3.1. The Overview of Enhanced Darkfield Hyperspectral Microscopy (EDHM)

EDHM is specifically designed to analyze a wide range of materials and biological samples in situ. The system utilizes tungsten halogen light (Dolan Jenner, Boxborough, MA, USA) as a source of illumination, which generates a spectral output from 400 to 2200 nm [25]. The light source has a variable power control output from 0–150 watts. In our experiment, the light source was set to a 150 watt output. The light is directed to the microscope via a liquid guide (Newport Inc., Newport, CA, USA), connected directly to the enhanced darkfield illuminator system. The enhanced darkfield illuminator system consists of an annular cardioid condenser, producing highly collimated light at oblique angles with a 1.2–1.4 NA.

Additionally, the darkfield illuminator contains collimating lenses. When the light guide is precisely connected to the enhanced darkfield illuminator, these collimating lenses modify the geometry of the source light to closely match the geometry of the system’s cardioid oil condenser. These collimating lenses also focus the light onto the condenser, which permanently refine and fix Koehler illumination of the light onto the condenser. This enables the highly oblique darkfield illumination to be focused onto the precise focal plane of the sample without losing the integrity of the Koehler illumination. This focusing of light is often referred to as critical illumination [26,27].

The darkfield illumination process generates images of samples with enhanced contrast and 10 times higher signal-to-noise ratio than conventional darkfield optics [25]. This indirect, oblique illumination upon interaction with the sample collects the reflected or elastically scattered light, permitting different visualization of objects with similar refractive indexes as the background [28]. These light scatters from the sample are then collected by a 60× oil iris objective (Olympus Inc., Tokyo, Japan). The optical image from the scattered light is projected onto a visible and near-infrared (VNIR) diffraction grating spectrograph (Specim, Oulu, Finland). This diffraction grating is a transmission-based grating that separates the light into distinct wavelengths from 400 nm to 1000 nm at high spectral resolution (~2 nm). The spectrally resolved light is then projected through a 30 µm slit onto the pixels of a charge-coupled device (CCD) video camera (PCO, Kelheim, Germany) [29].

The hyperspectral image is captured in a pixel row by pixel row line scan, and the obtained spectral information is the VNIR reflectance spectrum from the sample. An automated translational stage is utilized (Prior Scientific Instruments Ltd., Cambridge, UK) for capturing the hyperspectral image, which provides 10 nm step resolution. The stage and camera communicate with the hyperspectral image capture software to move the stage at precise distances to capture an image.

Once the hyperspectral image is captured, it is further analyzed using ENVI 4.8 hyperspectral image analysis software (Harris Geospatial Solutions, Inc., Herndon, VA, USA), which generates unique spectral signatures for individual pixels of the analyzed sample and saves it as RSL. The hyperspectral image stores the spectral data as three-dimensional datacubes, with X and Y dimensions being spatial and the Z dimension being spectral. The spectral response characteristic of the biological sample is subsequently mapped using SAM [25,30].

### 3.2. Propagation and Quantification of Human Coronaviruses, HCoV-OC43, and HCoV-229E

To demonstrate the applicability of EDHM for the differential detection of human coronaviruses, we acquired two commercially available strains, i.e., HCoV-OC43 and HCoV-229E, and used them to infect HCT-8 and MRC-5 cell lines at MOIs of 0.05 and 0.5, respectively [31]. The cytopathic effect (CPE) for HCoV-OC43 was detectable 11 days post-infection and involved the rounding-up and detachment of cells. The CPE for HCoV-229E-infected cells was observable following day 5 and involved cell spindling and detachment. Subsequently, the virions were isolated using freeze–thaw cycles performed in respective maintenance media followed by overnight precipitation in the presence of high molecular weight polyethylene glycol solution (PEG). High molecular weight PEG has been widely used for the isolation of many RNA viruses, including influenza virus [32], respiratory syncytial virus [33], and DNA viruses, e.g., bacteriophages [34], owing to its simplicity and capability to precipitate viruses at neutral pH and high ionic concentrations [35]. Following precipitation, the concentration of viral particles was assessed using plaque assays (Figure 2). The viral titers were calculated by dividing the average number of plaques by the dilution factor. For HCoV-OC43, the titer was estimated as 7.5 × 10^5^ PFU/mL, while for HCoV-229E, it was estimated as 6 × 10^5^ PFU/mL.

### 3.3. EDHM Analysis of HCoV-OC43 and HCoV-229E Virions

The hyperspectral images of three biological replicates of commercially obtained and prepared viral stocks of HCoV-OC43 and HCoV-229E were captured using EDHM with the numerical aperture of the 60x objective set 1.2 NA and hyperspectral camera exposure time set to 0.4 s for HCoV-OC43 and 0.7 s for HCoV-229E samples. The obtained hyperspectral images had the complete VNIR spectral data ranging from 400 to 1000 nm for each pixel with 2 nm spectral resolution across the full VNIR wavelength range. The VNIR 400 to 1000 nm range has been previously shown to provide the best combination of spectral and spatial resolution for detecting viral particles [36]. In addition, the use of enhanced darkfield light illumination optics improved the scatter properties of the virus particles, which contributed to optimum visualization of individual viral particles in the hyperspectral image.

The analysis of hyperspectral images of viral particles in solution was performed using the mean spectral analysis algorithm, which generates mean spectral responses for individual pixels. The mean spectra were normalized to a peak intensity of 1, and the spectral responses for the stock and precipitated HCoV-OC43, and HCoV-229E samples were compared within the same graph.

The analysis resulted in almost identical spectral profiles for the commercially acquired and prepared virions of HCoV-OC43 and HCoV-229E (Figure 3). However, the mean spectral comparisons of the HCoV-OC43 and HCoV-229E samples demonstrated the different spectral responses with ~30 nm shift of the spectral peaks. This is a crucial distinguishable feature that can be used for differential mapping and detecting both coronavirus strains in the solution. As these spectral plots represent the optical spectra derived in the process of converting the optical image from the scattered light to the spectral data, the resulting images frequently lose their original quality and the associated plots may not always generate sharp peaks with narrow full width at half maximum.

### 3.4. EDHM Analysis of HCT-8 and MRC-5 Infected Cell Lines

We have also performed EDHM analysis of uninfected and infected mammalian cells (HCT-8 and MRC-5) to assess whether we can identify the infection and differentiate HCoV strains within the cellular milieu. The images of three independent biological replicates of uninfected and infected HCT-8 and MRC-5 cells were captured using a hyperspectral camera with a detection range of 400 to 1000 nm, a 60x microscope objective magnification with a 6.4 µm camera pixel size and 2x pixel binning. In the resulting images, each pixel consists of the spectral profile of the sample at the corresponding spatial position and can be used to detect the viral particles with near-diffraction-limited resolution (1 μm). The RSL for the infected cells was created by filtering out the spectrum obtained for uninfected cells, which was regarded as a background using the Filter Spectral algorithm. Then, the RSLs were used to map hyperspectral images by employing the SAM function in the ENVI 4.8 software, which allows for the location and identification of viral particles within the infected cell. This filtering process ensured that the resulting spectral libraries reflected the actual presence of virions and were not artifacts.

The hyperspectral image analysis of HCT-8 cells infected with HCoV-OC43 resulted in a spectral peak of ~575 nm, in contrast to uninfected cells with a spectral peak of ~525 nm (Figure 4). In addition, we observed a significant difference in the optical spectral response for the mapped areas of HCoV-229E infected MRC-5 cells with a spectral peak at ~650 nm. In comparison, the total visual spectral response for uninfected MRC-5 cells had a broader range from ~500 nm to 620 nm, with the apex of the spectral plot at ~550 nm (Figure 5). Overall, the mean spectral profiles of uninfected and infected HCT-8 and MRC-5 cells revealed unique spectral mappings corresponding to viral particles detected within the cellular milieu. Thus, the EDHM is applicable to differentiate between infected and uninfected cells and between both HCoV strains.

The viral particles within the infected cell milieu (as in Figure 4b,c) were highly dispersed and scattered light less effectively than larger, brighter signals obtained for purified virion samples. Thus, detecting viruses in the infected cells required hyperspectral analysis. Also, as the hyperspectral images store the spectral data in the form of three-dimensional datacubes [25], the spectral mapping of viruses in the infected cells can sometimes allude to the localization of viruses inside the nucleus rather than in the cytoplasm (Figure 4c). In order to evaluate the subcellular distribution of virions, one can perform Z-stack image analysis. Further, the presence of viral particles outside the cells and their localization at the membranes was likely the result of cellular disintegration and virus budding [36].

## 4. Discussion

Recognition of the importance of community diseases caused by human coronaviruses has increased in recent years; however, detailed information on pathogenesis, immunity, and viral characteristics remains limited. As such, significant efforts have been made to develop more sensitive diagnostic tools and molecular detection methods [4]. In this manuscript, we described the application of enhanced darkfield hyperspectral microscopy (EDHM), a label-free technique for the direct visualization and relative quantitative analysis of two human coronaviruses, i.e., HCoV-OC43 and HCoV-229E, present in a solution and within a cellular milieu.

We mapped the commercially available and prepared virions for both coronavirus strains using the hyperspectral image analysis. We found that their spectral profiles were identical, and as such, we concluded that the source of virions does not impact their spectral signal. Subsequently, we compared the mean spectral profiles of HCoV-OC43 and HCoV-229E virions and noted distinct spectral signals characterizing these viral strains. Further, the spectral profiles of uninfected and infected cells were compared to understand the extent to which these signatures can be used to identify the presence of virions and their differentiation. Here, we noted that the spectral signatures of the infected cells exhibited a considerable peak shift to a higher wavelength compared to uninfected cells. Also, we noticed that the spectral signatures of HCoV-OC43 and HCoV-229E in the infected cells were distinct. These unique spectral responses from the virions can be utilized to detect these viral strains.

EDHM offers a significant advantage over conventional imaging techniques, involving non-destructive sample preparation, fast image acquisition, and rapid analysis. Additionally, it can determine samples’ spatial and spectral information in a single measurement by storing the hyperspectral image in three-dimensional datacubes [37]. Moreover, EDHM can be operated by a relatively inexperienced individual with minimal training compared to other microscopic techniques, such as transmission electron microscopy (TEM) or scanning electron microscopy (SEM). The EDHM system is also significantly less expensive than alternative options, with estimated costs for the system averaging approximately $155,000. In comparison, the TEM system costs an average of $4.0 million, while the SEM system expenses are nearly $1.0 million [24]. Although the EDHM confers various advantages over conventional microscopy techniques, in terms of cost and time reduction, the amount of data generated during the analysis requires substantial processing time, ample data storage, and accurate analysis to extract conclusive information [38,39]. Occasionally, along with the reflected/scattered light from the sample, the out-of-focus light also reaches the objective, reducing the spatial resolution of the image. Therefore, appropriate instrumental adjustments are required to provide the users with vertical scanning capabilities [40]. These aspects should be considered during the analysis.

EDHM has been applied to deliver real-time images of biomarker information and examine cell pathophysiology based on the spectral resonance characteristics of specific tissues. For example, EDHM has been used for in vitro screening of chemical entities for amyloidogenesis modulatory activity and the detection of Aβ aggregate in Alzheimer’s mouse brain and retina [41]. EDHM is also a convenient, non-invasive tool for assessing signs of hemorrhagic shock (HEM), where the quantification of changes in the surface tissue saturation of oxygen (S_HSI_O_2_), estimated through EDHM, is reflective of skin color changes and mottling during HEM [42]. Furthermore, EDHM, in combination with an artificial neural network (ANN), has been used for the diagnosis of urolithiasis recidivism [43,44]. In this case, EDHM provides rapid characterization and classification of renal calculi within the urinary tract compared to conventional techniques like stereoscopic microscopy [45] and infrared analysis [46] which require laborious methodology and sample pretreatment. EDHM has also been used to evaluate the unique spectral profiles and cellular localization of single-walled carbon nanotubes (SWCNT) and the influenza A virus (IAV) particles in fixed small airway epithelial cells [36]. Here, the spectral analysis allowed to the conclusion that the coexposure with SWCNT lead to increased viral accumulation within the cell.

Recently EDHM has been applied to track gold nanoparticles capped with antisense oligonucleotides used as biomarkers against SARS-CoV-2. In the presence of SARS-CoV2 infection, a significant number of agglomerated gold nanoparticles were detected, indicating their specific binding to SARS-CoV-2 RNA. Further, a significant hyperspectral shift and broadening of hyperspectral signatures were observed for the nanoparticles in the presence of the viral RNA [47]. Also, EDHM has been used for hyperspectral mapping of metal oxide nanoparticles in histological and aqueous samples [48,49]. A modified version of the EDHM system, including the outlier removal auxiliary classifier generative adversarial nets (OR-AC-GAN), has been applied to detect early symptoms of the disease caused by Tomato Spotted Wilt Virus (TSWV). OR-AC-GAN is a neural network architecture in the deep learning domain [50,51,52] that, in combination with EDHM, has been utilized for image segmentation, feature extraction, and spectrum classification. This modified EDHM system can distinguish the pixels of healthy and TSWV-infected plants at early stages before the symptoms are visible, thus facilitating the management and spread of disease [53]. Certainly, combining the capacity of EDHM with recent advances in machine learning will enable new diagnostics and medical imaging to find, classify, and interpret relevant phenomena [54].

In summary, the EDHM represents a novel diagnostic tool that can be utilized to detect and differentiate infectious agents, including human coronaviruses, present in biological and environmental samples, i.e., in the solution and within infected mammalian cells. Considering numerous publications reporting the presence of coronavirus virions in water, wastewater, and sewage [55,56], this method can aid in wastewater-based epidemiology and sewage surveillance, providing valuable insights into the prevalence of viruses among the human population and could serve as a sensitive surveillance system and a crucial early warning tool. Additionally, combining EDHM with other techniques like Raman spectroscopy, Fourier-transformed infrared can improve the diagnostic applications of this technique by enhancing the resolution and penetration depth within the heterogenous tissue and cellular environment.

## Figures and Tables

**Figure 1 jimaging-08-00024-f001:**
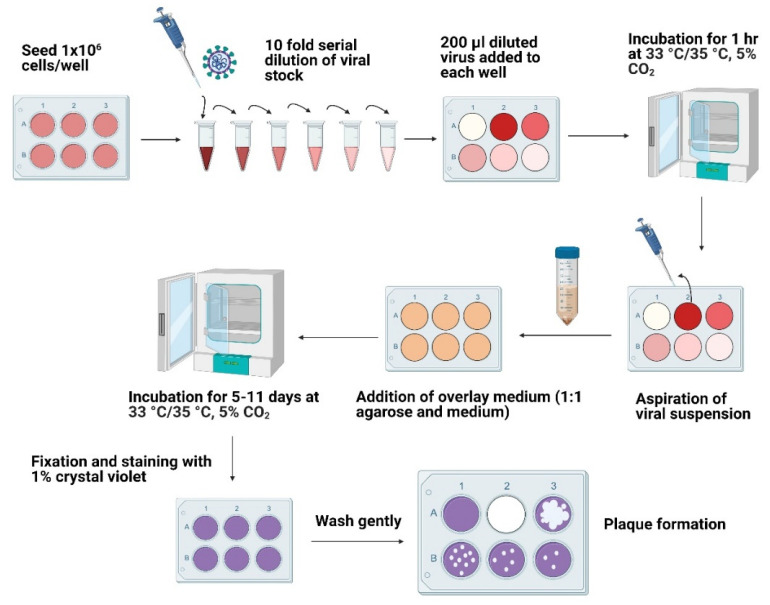
Schematic representation of the plaque assay procedure. The cells were grown to near full confluency in a 6-well plate. The viral stocks were serially diluted, added to each well, and incubated for 1 h at 33 °C for HCoV-OC43 and 35 °C for HCoV-229E in 5% CO_2_. The viral suspensions were aspirated, and overlay media was added to each well. The plates were incubated for 11 days at 33 °C in 5% CO_2_ for HCoV-OC43 infected cells or 5 days at 35 °C in 5% CO_2_ for HCoV-229E infected cells. The cells were fixed and stained with 1% crystal violet post-infection followed by plaque counting.

**Figure 2 jimaging-08-00024-f002:**
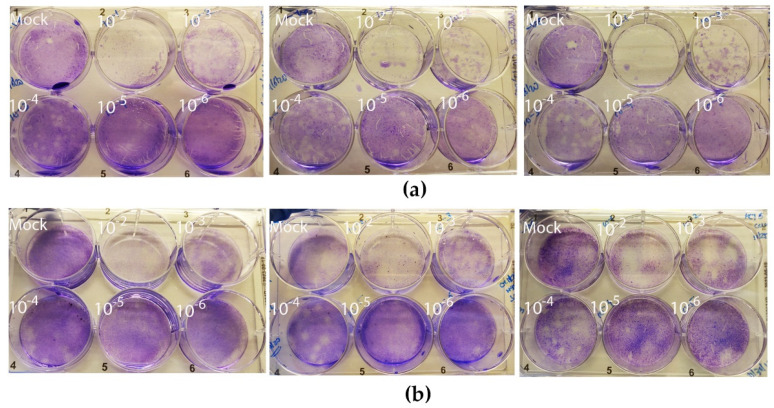
Plaque assay for assessment of HCoV viral titers. Plaques were observed at 11 days post-infection (dpi) with (**a**) HCoV-OC43 on a monolayer of HCT-8 and (**b**) 5 dpi with HCoV-229E on a monolayer of MRC-5. All the assays were performed in triplicates in a 6-well plate with a 10-fold serial dilution of viral stocks (values indicated on top). Mock infected cells were included as controls.

**Figure 3 jimaging-08-00024-f003:**
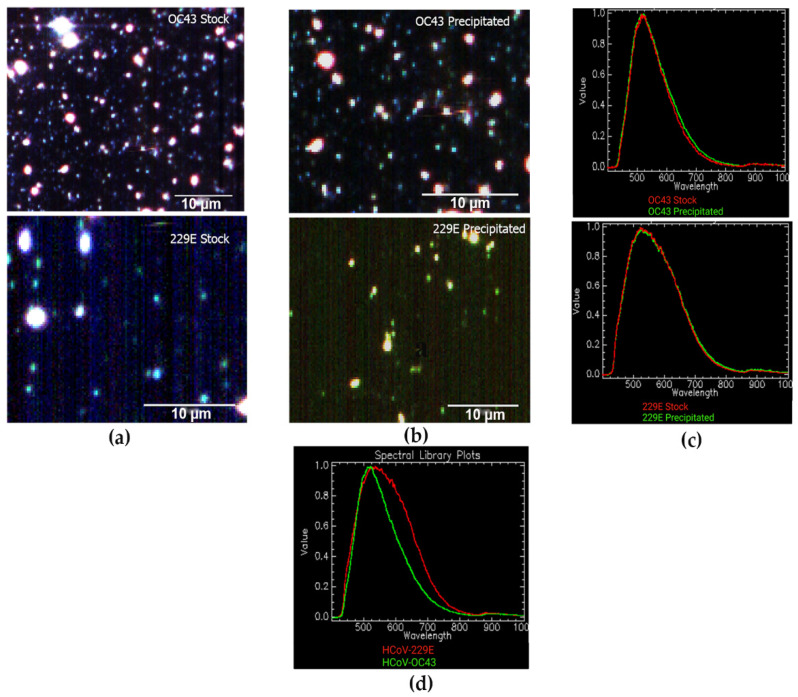
(**a**) EDHM imaging of commercially acquired viral stock of HCoV-OC43 (top) and HCoV-229E (bottom) viewed under 60x oil iris objective; (**b**) EDHM imaging of prepared viral stocks of HCoV-OC43 (top) and HCoV-229E (bottom). The white spots correspond to the viral particles. (**c**) The overlapping spectral profiles for HCoV-OC43 (top) and HCoV-229E (bottom) obtained commercially (red) and prepared by PEG precipitation (green). (**d**) Hyperspectral signal curves generated for HCoV-OC43 (green) and HCoV-229E (red) viral solution indicate a spectral peak difference of ~30 nm.

**Figure 4 jimaging-08-00024-f004:**
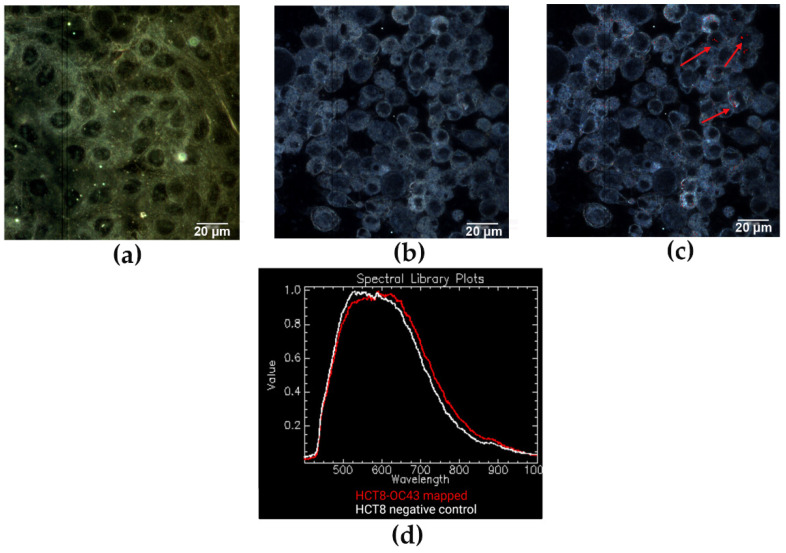
(**a**) EDHM imaging of uninfected and (**b**) infected HCT-8 cells visualized under 60x oil iris objective. (**c**) Spectral mapping resulted in the identification of pixels (red dots and indicating arrows) corresponding to HCoV-OC43 viral particles. (**d**) Hyperspectral signal curves obtained for HCT8 uninfected (white, negative control), and infected (red) cells show a peak shift of ~50 nm.

**Figure 5 jimaging-08-00024-f005:**
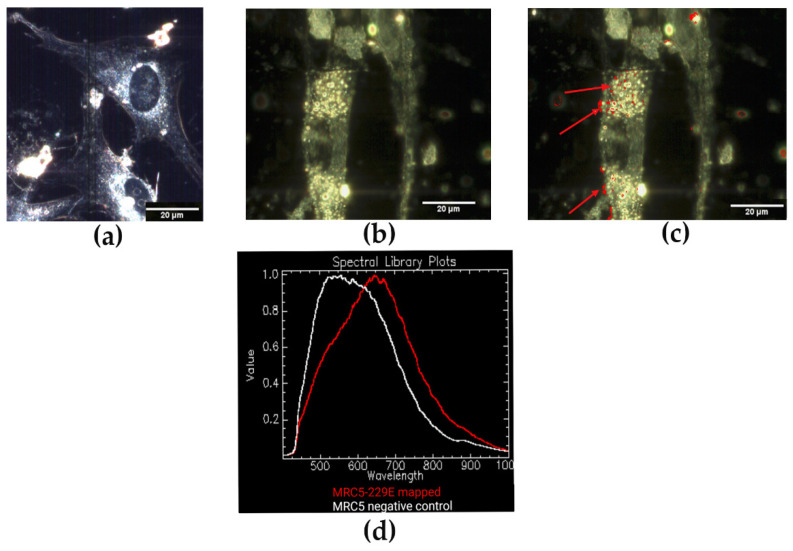
EDHM imaging of (**a**) uninfected and (**b**) infected MRC-5 cells visualized under 60x oil iris objective. (**c**) Spectral mapping of HCoV-229E virus pixels in the infected cells are indicated by red dots (red arrows). (**d**) Hyperspectral signal curves corresponding to uninfected (white), and infected (red) MRC-5 cells show a peak shift of ~100 nm.

## Data Availability

Not applicable.

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
