# Peer review of "Label-Free Detection of Human Coronaviruses in Infected Cells Using Enhanced Darkfield Hyperspectral Microscopy (EDHM)"

_2313-433X, 2022, doi:10.3390/jimaging8020024_

Round 1

Reviewer 1 Report

In this manuscript, Gosavi et al. describe a microscopy technique (developed by CytoViva Inc.) based on enhanced dark field microscopy (EDHM) to identify human coronavirus particles in infected cells. The main advantage of this technique is the label-free and non-laborious detection of the particles.

To my current knowledge, there is no scientific literature about EDHM being used to detect viral particles. Therefore, there is a strong component of novelty in this manuscript. However, I have some serious concerns about the results shown in this manuscript.

Introduction,

Lines 76-79: this comparison is not fair. First, single-virus tracking requires live cell imaging while the technique that authors describe here is based on fixed cells. Second, photobleaching might be an issue sometimes but most commercially available fluorophores conjugated to secondary antibodies have great photostability that can be even improved using reagents such as ProLong Gold glass mountant. Third, fluorescence microscopy not only allows visualization of viruses, but also is the one the main technique when studying virus-host interactions at molecular level.

Authors should reconsider this statement.

Materials and methods

Lines 114 and 115: about the term "transfected", here the authors refer to cells exposed to a certain viral titer, I think authors must refer to "infection" instead.

Lines 185:  a small typo on "µL"

Results

My biggest concerns here are related to EDHM imaging of infected cells. In Figure 4c and 5c, the authors indicate the HCoV positions using red pixels. When checking the same EDHM pictures from infected cells in Figures 4b and 5b, the position of those red pixels do not correspond with bright pixels. According to the authors, viral particles should be detected by EDHM as shown in Figure 3. Why viral particles were not visible in infected cells?

Moreover, the position of the red pixels in Figure 4c seem to be located inside the nucleus. The replication cycle of coronaviruses involves a series of steps in the cell cytoplasm. How can authors explain the presence of viral coronavirus particles inside the nucleus?

Also, Figure 5c shows an uncommon distribution of virus outside of the cell: either bound to the membrane or in some extracellular vesicles that could be an artifact of the fixation process.

Authors need to demonstrate, using other microscopy techniques, that the spots shown here specifically correspond to HCoV particles.

Minor: picture quality should be increased.

Author Response

Reviewer 1

In this manuscript, Gosavi et al. describe a microscopy technique (developed by CytoViva Inc.) based on enhanced dark field microscopy (EDHM) to identify human coronavirus particles in infected cells. The main advantage of this technique is the label-free and non-laborious detection of the particles. To my current knowledge, there is no scientific literature about EDHM being used to detect viral particles. Therefore, there is a strong component of novelty in this manuscript. However, I have some serious concerns about the results shown in this manuscript.

We thank the reviewer for this very insightful and constructive critique of our manuscript. Below, we have outlined a point-by-point response to the reviewer’s suggestions and made changes and adjustments through the ‘track changes’ function in the word file.

Introduction,

Lines 76-79: this comparison is not fair. First, single-virus tracking requires live-cell imaging, while the technique that the authors describe here is based on fixed cells. Second, photobleaching might be an issue sometimes but most commercially available fluorophores conjugated to secondary antibodies have great photostability that can be even improved using reagents such as ProLong Gold glass mountant. Third, fluorescence microscopy not only allows visualization of viruses but also is one the main technique when studying virus-host interactions at the molecular level.

The authors should reconsider this statement.

Response: Following the reviewer’s suggestion, we have modified the text in the ‘Introduction’ section (lines 76-82):

“Fluorescence microscopy is another widely employed detection method. It allows visualization of viral particles and is one of the main techniques to study molecular aspects of viral infectivity cycle and virus interactions within the complex biological environment in a minimally invasive manner [20]. However, sample preparation and analysis are time-consuming as they require the labeling of viruses with fluorescent dyes [21]. Also, endogenous cellular autofluorescence can mask a proper fluorescence signal, limiting the ability to visualize low abundance fluorescently labeled virions [22]”

Materials and methods

Lines 114 and 115: about the term “transfected”, here the authors refer to cells exposed to a certain viral titer, I think authors must refer to “infection” instead.

Response: As per the suggestion of the reviewer, we have replaced  the word “transfected” with “infected” in (line 115).

Lines 185:  a small typo on “µL.”

Response: We thank the reviewer for pointing out the typo, which now has been fixed in (line 183).

Results

My biggest concerns here are related to EDHM imaging of infected cells. In Figure 4c and 5c, the authors indicate the HCoV positions using red pixels. When checking the same EDHM pictures from infected cells in Figures 4b and 5b, the position of those red pixels does not correspond with bright pixels. According to the authors, viral particles should be detected by EDHM as shown in Figure 3. Why viral particles were not visible in infected cells?

Response: We thank the reviewer for this insightful comment. We have included the explanation of this effect under the ‘Result’ section, under ‘EDHM analysis of HCT-8 and MRC-5 infected cell lines’ paragraph,  (lines 358-361). Please see below:

“The viral particles within the infected cell milieu (as in Figures 4b and 4c) were highly dispersed and scattered light less effectively than larger, brighter signals obtained for purified virion samples. Thus, detecting viruses in the infected cell required further hyperspectral analysis.”

Moreover, the position of the red pixels in Figure 4c seems to be located inside the nucleus. The replication cycle of coronaviruses involves a series of steps in the cell cytoplasm. How can authors explain the presence of viral coronavirus particles inside the nucleus?

Response: We thank the reviewer for this comment. We have addressed this issue in the ‘Results’ section, under ‘EDHM analysis of HCT-8 and MRC-5 infected cell lines’ paragraph, (lines 361-365):

“Also, as the hyperspectral images store the spectral data in the form of three-dimensional datacubes [25], the spectral mapping of viruses in the infected cells can sometimes allude to the localization of viruses inside the nucleus rather than in the cytoplasm (Figure 4c). In order to evaluate the subcellular distribution of virions, one can perform Z-stack image analysis.”

Also, Figure 5c shows an uncommon distribution of virus outside of the cell: either bound to the membrane or in some extracellular vesicles that could be an artifact of the fixation process.

Response: We have included the following statement under the ‘Results’ section, ‘EDHM analysis of HCT-8 and MRC-5 infected cell lines’ paragraph, (lines 365-367) to address this issue.

“Further, the presence of viral particles outside the cells and their localization at the membranes was likely the result of cellular disintegration and virus budding [36].”

Authors need to demonstrate, using other microscopy techniques, that the spots shown here specifically correspond to HCoV particles.

Response: We appreciate this valuable suggestion. EDHM has been previously employed as a stand-alone method, e.g., used for differentiating the blood cells, mapping cellular components within erythrocytes, evaluate gold nanoparticle distribution within Daphnia magna.  [1][2][3]

  1. Conti, M.; Scanferlato, R.; Louka, M.; Sansone, A.; Marzetti, C.; Ferreri, C. Building up spectral libraries for mapping erythrocytes by hyperspectral dark field microscopy. Biomed. Spectrosc. Imaging 2016, 5, 175–184, doi:10.3233/bsi-160133.
  2. Sacco Verebes, G.; Melchiorre, M.; Garcia-Leis, A.; Ferreri, C.; Marzetti, C.; Torreggiani, A. Hyperspectral enhanced dark field microscopy for imaging blood cells., doi:10.1002/jbio.201300067.
  3. Botha, T.L.; Boodhia, K.; Wepener, V. Adsorption, uptake and distribution of gold nanoparticles in Daphnia magna following long term exposure. Aquat. Toxicol. 2016, 170, 104–111, doi:10.1016/J.AQUATOX.2015.11.022.

Thus, we feel that EDHM itself is sufficient for the quick and robust detection of viral particles. To assure the accuracy of our spectral mapping, we filtered the spectra obtained from virions within infected cells with the spectra acquired from uninfected cells. This verified that the hyperspectral mapping was related to the virions and was not artifacts.

This has now been indicted in the manuscript in ‘The Materials and Methods’ section, ‘EDHM imaging of mammalian cells infected with HCoV’ paragraph, (lines 220-230):

“For hyperspectral mapping analysis, the infected cells were scanned for large areas of individual pixel level spectrum and were grouped in regions of interest (ROIs). Each of these ROIs were then converted to spectral libraries (SLs) representing thousands of individual pixel spectral data obtained from each pixel of ROI. Individual SLs obtained from the infected cells were then filtered against the SLs acquired from the uninfected cells, using the Filter Spectral Library algorithm to remove duplicative spectra and create a  Reference Spectral Library (RSL). The RSL was then used to map against all the data cubes obtained from the infected cells for detecting the areas with the same spectral profile as the RSL using the SAM function in the ENVI 4.8 software. The mapped areas were merged in one color and overlaid on the original data cube to observe where the viral particles of interest were located. “

Minor: picture quality should be increased.

Response: We thank the reviewer for raising this issue. To respond to the reviewer’s concern, we have added an explanation regarding the quality of the picture in the ‘Results' section, ‘EDHM analysis of HCoV-OC43 and HCoV-229E virions’ paragraph, lines (321-324):

“As these spectral plots represent the optical spectra derived in the process of converting the optical image from the scattered light to the spectral data, the resulting images frequently lose their original quality, and the associated plots may not always generate sharp peaks with a narrow full width at half maximum.”

Also, we have included the image files and hyperspectral plots with an improved resolution.

Reviewer 2 Report

In the manuscript ' Label-free detection of human coronaviruses in infected cells using Enhanced Darkfield Hyperspectral Microscopy,' the authors Gosavi et al reported their research on the dark-field imaging in the label-free detection of two 84 HCoV strains in infected mammalian cells which is very attractive technique having quite clinical application in detecting coronaviruses in human cells. I suggest this manuscript accepted as the publication of Journal of Imaging after the following question can be answered.

  1. The setup of the imaging system is missed, and the authors are encouraged to show it.
  2. The lipid is more sensitive at 1730nm, why the author chose 650nm?

Author Response

Reviewer 2

In the manuscript ' Label-free detection of human coronaviruses in infected cells using Enhanced Darkfield Hyperspectral Microscopy,' the authors Gosavi et al reported their research on the dark-field imaging in the label-free detection of two 84 HCoV strains in infected mammalian cells which is a very attractive technique having quite clinical application in detecting coronaviruses in human cells. I suggest this manuscript be accepted as the publication of Journal of Imaging after the following question can be answered.

We would like to thank the reviewer for the thoughtful comments and efforts towards improving our manuscript. We have revised the manuscript and implemented the changes, which are indicated in the manuscript through the 'track changes' function in the word file.

  1. The setup of the imaging system is missed, and the authors are encouraged to show it.

Response: We appreciate the reviewer’s comment on this. Now, the manuscript includes an in-depth description of the system set up in the 'Results' section, 'The overview of enhanced darkfield hyperspectral microscopy (EDHM),' (lines 236-272):

“The light source has a variable power control output from 0 – 150 watts. In our experiment, the light source was set to 150-watt output. The light is directed to the microscope via a liquid guide (Newport Inc., Newport, CA), connected directly to the enhanced darkfield illuminator system. The enhanced darkfield illuminator system consists of an annular cardioid condenser, producing highly collimated light at oblique angles with a 1.2-1.4 NA.

Additionally, the darkfield illuminator contains collimating lenses. When the light guide is precisely connected to the enhanced darkfield illuminator, these collimating lenses modify the geometry of the source light to closely match the geometry of the system’s cardioid oil condenser. These collimating lenses also focus the light onto the condenser, which permanently refines and fix Koehler illumination of the light onto the condenser. This enables the highly oblique darkfield illumination to be focused onto the precise focal plane of the sample without losing the integrity of the Koehler illumination. This focusing of light is often referred to as critical illumination [26][27].

The darkfield illumination process generates images of samples with enhanced contrast and a 10 times higher signal-to-noise ratio than conventional darkfield optics [25]. This indirect, oblique illumination upon interaction with the sample collects the reflected or elastically scattered light, permitting different visualization of objects with similar refractive indexes as the background [28]. These light scatters from the sample are then collected by a 60x oil iris objective (Olympus Inc., Tokyo, Japan). The optical image from the scattered light is projected onto a visible and near-infrared (VNIR) diffraction grating spectrograph (Specim, Oulu, Finland). This diffraction grating is a transmission-based grating that separates the light into distinct wavelengths from 400 nm - 1,000 nm at high spectral resolution (~2 nm). The spectrally resolved light is then projected through a 30 µm slit onto the pixels of a charge-coupled device (CCD) video camera (PCO, Kelheim, Germany) [29].

The hyperspectral image is captured in a pixel row by pixel row line scan, and the obtained spectral information is the VNIR reflectance spectrum from the sample. An automated translational stage is utilized (Prior Scientific Instruments Ltd, Cambridge, UK) for capturing the hyperspectral image, which provides 10 nm step resolution. The stage and camera communicate with the hyperspectral image capture software to move the stage at precise distances to capture an image.

Once the hyperspectral image is captured, it is further analyzed using ENVI 4.8 hyperspectral image analysis software (Harris Geospatial Solutions, Inc., Herndon, VA), which generates unique spectral signatures for individual pixels of the analyzed sample and saves it as RSL. The hyperspectral image stores the spectral data as three-dimensional datacubes, with X and Y dimensions being spatial and the Z dimension being spectral.”

  1. The lipid is more sensitive at 1730nm, why did the author choose 650nm?

Response: We thank the reviewer for this insightful comment. We have clarified this in the ‘Results’ section, ‘EDHM analysis of HCoV-OC43 and HCoV-229E virions ' paragraph, (lines 303-307):

“The obtained hyperspectral images had the complete VNIR spectral data ranging from 400 - 1,000 nm for each pixel with 2 nm spectral resolution across the full VNIR wavelength range. The VNIR 400 to 1000 nm range has been previously shown to provide the best combination of spectral and spatial resolution for detecting viral particles [36].”

 Sanpui, P.; Zheng, X.; Loeb, J.C.; Bisesi, J.H.; Khan, I.A.; Afrooz, N.R.M.N.; Liu, K.; Badireddy, R.R.; Wiesner, M.R.; Ferguson, P.L.; et al. Single-walled carbon nanotubes increase pandemic influenza A H1N1 virus infectivity of lung epithelial cells. Part. Fibre Toxicol. 2014, 11, 1–15, doi:10.1186/s12989-014-0066-0.

Reviewer 3 Report

The author demonstrated using enhanced darkfield hyperspectral microscopy (EDHM) as a non-invasive and label-free tool for imaging biological samples. The technique can reach a spatial resolution of about 1 µm and a spectral resolution of 2 nm. Peak wavelength shift is observed among different viral stocks and between infected and uninfected cells, which is of significant scientific and clinical interests. I suggest the work to be accepted for publication after addressing the following comments.

  1. Is the spectral information arising from the uv-vis absorption of the sample?
  2. In line 321-322, the author stated that the spectra peak position is 500 nm, which is not accurate enough.
  3. The spectral plots in Fig. 3c-d, 4d, and 5d are not clear. Please replot.

Author Response

Reviewer 3

The author demonstrated using enhanced darkfield hyperspectral microscopy (EDHM) as a non-invasive and label-free tool for imaging biological samples. The technique can reach a spatial resolution of about 1 µm and a spectral resolution of 2 nm. Peak wavelength shift is observed among different viral stocks and between infected and uninfected cells, which is of significant scientific and clinical interest. I suggest the work be accepted for publication after addressing the following comments.

We appreciate the reviewer's positive feedback about our manuscript. Below, we have addressed the concerns and made changes through the word file's 'track changes' function.

  1. Is the spectral information arising from the UV-vis absorption of the sample?

Response: This has been more thoroughly addressed in the 'Results’ section, ‘EDHM analysis of HCoV-OC43 and HCoV-229E virions ' paragraph, (lines 303-307):

“The obtained hyperspectral images had the complete VNIR spectral data ranging from 400 - 1,000 nm for each pixel with 2 nm spectral resolution across the full VNIR wavelength range. The VNIR 400 to 1000 nm range has been previously shown to provide the best combination of spectral and spatial resolution for detecting viral particles [36].”

Sanpui, P.; Zheng, X.; Loeb, J.C.; Bisesi, J.H.; Khan, I.A.; Afrooz, N.R.M.N.; Liu, K.; Badireddy, R.R.; Wiesner, M.R.; Ferguson, P.L.; et al. Single-walled carbon nanotubes increase pandemic influenza A H1N1 virus infectivity of lung epithelial cells. Part. Fibre Toxicol. 2014, 11, 1–15, doi:10.1186/s12989-014-0066-0.

  1. In line 321-322, the author stated that the spectra peak position is 500 nm, which is not accurate enough.

Response: We thank the reviewer for catching this inaccuracy. We have modified the statement as follows in the ‘Results’ section, ‘EDHM analysis of HCT-8 and MRC-5 infected cell lines’ paragraph, (lines 350-354):

“In addition, we observed a significant difference in the optical spectral response for the mapped areas of HCoV-229E infected MRC-5 cells with a spectral peak at ~650 nm. In comparison, the total visual spectral response for uninfected MRC-5 cells had a broader range from ~500 nm - 620 nm, with the apex of the spectral plot at ~550 nm (Figure 5).”

  1. The spectral plots in Fig. 3c-d, 4d, and 5d are not clear. Please replot.

Response: We have addressed this in the ‘Result’ section of the manuscript, ‘EDHM analysis of HCoV-OC43 and HCoV-229E virions’ paragraph, (lines 321-324):

“As these spectral plots represent the optical spectra derived in the process of converting the optical image from the scattered light to the spectral data, the resulting images frequently lose their original quality, and the associated plots may not always generate sharp peaks with a narrow full width at half maximum.”

Also, we have included image files of the hyperspectral plots with an improved resolution.

Round 2

Reviewer 1 Report

Authors properly adressed all my concerns. The overall quality of the manuscript has increased.